# Topological phonon transport in an optomechanical system

Hengjiang Ren [1,2,6,7,9], Tirth Shah [3,4,9], Hannes Pfeifer [3,8], Christian Brendel[3], Vittorio Peano [3], Florian Marquardt [3,4] & Oskar Painter [1,2,5✉]

Light is a powerful tool for controlling mechanical motion, as shown by numerous applications in the field of cavity optomechanics. Recently, small scale optomechanical circuits, connecting a few optical and mechanical modes, have been demonstrated in an ongoing push towards multi-mode on-chip optomechanical systems. An ambitious goal driving this trend is to produce topologically protected phonon transport. Once realized, this will unlock the full toolbox of optomechanics for investigations of topological phononics. Here, we report the realization of topological phonon transport in an optomechanical device. Our experiment is based on an innovative multiscale optomechanical crystal design and allows for site-resolved measurements in an array of more than 800 cavities. The sensitivity inherent in our optomechanical read-out allowed us to detect thermal fluctuations traveling along topological edge channels. This represents a major step forward in an ongoing effort to downscale mechanical topological systems.

[1] Thomas J. Watson, Sr., Laboratory of Applied Physics and Kavli Nanoscience Institute, California Institute of Technology, Pasadena, CA 91125, USA. [2] Institute for Quantum Information and Matter, California Institute of Technology, Pasadena, CA 91125, USA. [3] Max Planck Institute for the Science of Light, Staudtstr. 2, 91058 Erlangen, Germany. [4] Department of Physics, Friedrich-Alexander Universität Erlangen-Nürnberg, Staudtstr. 7, 91058 Erlangen, Germany. [5] AWS Center for Quantum Computing, Pasadena, CA 91125, USA. [6] Present address: Institute of High Performance Computing, Agency for Science, Technology and Research (A*STAR), Singapore 138632, Singapore. [7] Present address: Anyon Computing Inc, Dover, DE 19901, USA. [8] Present address: Institut für Angewandte Physik, Universität Bonn, Wegelerstr. 8, 53115 Bonn, Germany. [9] These authors contributed equally: Hengjiang Ren, Tirth Shah. ✉email: opainter@caltech.edu

Recent advances in cavity optomechanics[1] have now made it possible to use light not just only as a passive measuring device of mechanical motion[2] but also to manipulate the motion of mechanical objects down to the level of individual quanta of vibrations (phonons). At the same time, micro-fabrication techniques have enabled small-scale optomechanical circuits capable of on-chip manipulation of mechanical and optical signals[3–12]. Building on these developments, theoretical proposals have shown that larger-scale optomechanical arrays can be used to modify the propagation of phonons, realizing a form of topologically protected phonon transport[12–16]. This optomechanical approach is part of a broader endeavor to realize topological mechanical devices[17] in a variety of platforms ranging from pendula[18,19], to sound waves in fluids[20,21], and vibrations in solids[22–26], exploiting the general concepts of topologically robust wave propagation[27]. It is motivated by the quest to reduce the footprint of mechanical topological devices towards systems at the nanoscale, where hypersonic frequency ($\gtrsim$GHz) acoustic wave circuits consisting of robust delay lines[28] and non-reciprocal elements[29–32] may be implemented. Owing to the broadband character of the topological channels, the control of the flow of heat-carrying phonons may also be envisioned.

Here, we report the observation of topological phonon transport in an optomechanical device comprising over 800 cavity-optomechanical elements at room temperature fabricated on the surface of a silicon microchip. Using sensitive, spatially resolved optical read-out[33,34] we detect thermal phonons traveling along a topological edge channel. We observe a substantial reduction of backscattering in a 0.325–0.34 GHz band demonstrating unprecedented carrier frequency and bandwidth compared to the only existing nanoscale on-chip topological mechanics implementations[25,26]. A key innovation of our work is to introduce a design paradigm for optomechanical devices based on multiscale optomechanical crystals (OMCs). Standard single-scale OMCs[35–38] are patterned free-standing structures that can be engineered to yield large radiation-pressure coupling between cavity photons and phonons with similar wavelengths. In contrast, our multiscale device consists of a superlattice structure, superimposing two patterns with very different but commensurate lattice spacings. This multiscale approach adds an extra degree of flexibility, decoupling the engineering of photonic and phononic modes. In our design, the larger scale defines a phononic crystal. Embedded within each unit cell of the phononic crystal is a smaller scale photonic crystal, which hosts a high-Q optical nanocavity for optical site-resolved read-out of phonons. Local changes within the OMC lattice of the phononic crystal unit cell are used to create topologically distinct mechanical domains, whose interface hosts phononic helical edge states based on the Valley Hall effect[39,40].

## Results

### Design of the multiscale OMC for topological phononics.

Images of a fabricated multiscale OMC structure are shown in Fig. 1a, b (see Supplementary Note 2 for more details on device fabrication). In our design, a triangular lattice of snowflake-shaped holes with lattice spacing $a_m = 16.02\ \mu m$ is superimposed onto another triangular lattice of cylindrical holes with a much smaller spacing $a_o = 450\ nm$. This hole pattern has been etched into the thin (220 nm thickness) silicon device layer of a silicon-on-insulator (SOI) microchip. After releasing the underlying buried oxide layer, this produces an array of connected triangular silicon membranes forming the phononic crystal, each hosting a photonic crystal defined by the smaller holes (see Fig. 1b and inset). The snowflake pattern is adopted from a well-known single-scale OMC design[37,38] and has also been proposed theoretically as a platform for topological phononics[14,15]. In this work we have increased the snowflake lattice spacing by a factor of ~30,

enabling every triangular membrane to harbor an optical nano-cavity consisting of a localized defect in the triangular photonic-crystal hole pattern. The purpose of using a cavity is to boost the optomechanical interaction (see Supplementary Note 3 and Supplementary Fig. 1). In Fig. 1a, b, such a cavity is present only in the downward-pointing triangular membranes, with the upward-pointing triangular membranes having an unperturbed photonic-crystal pattern. Although the two lattices (phononic and photonic) are at vastly different scales, the patterning of the photonic crystal within each triangular membrane does (weakly) influence the phononic properties, providing an extra knob to trim the mechanical properties.

We employ these tuning knobs of the multiscale design to realize a structure supporting robust helical edge states based on the Valley Hall effect[39]. The Valley Hall effect is relevant for a wide range of systems that support Dirac cones, including electronic[39,40], photonic[30], and mechanical systems[20,23]. In this context, valley refers to the quasi-momentum region around a Dirac cone. In a time-reversal-symmetric system, the Dirac cones, and thus the corresponding valleys, come in pairs mapped onto each other by the operation of time reversal. Thus, the valley can be viewed as a binary degree of freedom akin to the spin. In the Valley Hall effect, valley-polarized edge excitations propagate in opposite directions, analogous to spin-polarized edge states in the Spin Hall effect.

As we are pursuing an optomechanical approach to the detection of mechanical edge excitations, we focus here on the vibrational modes that couple to light, the in-plane modes which are even under the mirror operator $M_z$ ($z \mapsto -z$). For these modes, the snowflake phononic crystal supports a pair of Dirac cones well-isolated from the remaining bands[14,15]. In our experiment, the Dirac cones have a center frequency of ~0.3 GHz, with linear Dirac-like dispersion across a bandwidth of 70 MHz (see Fig. 1c). These cones are protected by a symmetry under $M_y$ (see Supplementary Note 4). We open the bulk bandgap that will host the helical edge states by breaking this symmetry. Decreasing the size of the photonic-crystal holes in the upward-pointing triangles by a factor of 0.78 produces a bandgap of width 18 MHz (see Fig 1d). The underlying vibrational Bloch waves, calculated using finite-element method (FEM) simulations (see Supplementary Note 1), are shown in Fig. 1e, f. A comparatively large unit-cell vacuum optomechanical coupling ($g_0 = 2\pi \times 33.7$ kHz) is produced for the higher-frequency mode in Fig. 1f because it displays breathing motion around the optical cavity. A detailed discussion of the optomechanical coupling is provided in Supplementary Notes 3, 9.

In the Valley Hall effect, the topological transport takes place through counter-propagating valley-polarized edge states which exist at the domain walls separating two topologically distinct domains of opposite so-called valley Chern number. By applying the mirror operation $M_y$, we construct from the deliberately mirror-symmetry-broken design described above a second domain with opposite valley Chern numbers (see Fig. 1g, h). The key feature leading to robust transport is that edge excitations can navigate a path with arbitrarily sharp angles while still remaining confined within the same valley region of quasi-momentum space. On the other hand, backscattering would require a large quasi-momentum transfer to reach a different valley, and is thus strongly suppressed. Our fit to the Dirac Hamiltonian describing our anisotropic structure (see Methods) shows both a dependence of the bandstructure on the domain wall orientation and some deviations from the idealized theoretical limiting case. For a horizontal domain wall, this leads to in-gap edge states that extend only through part of the full bandgap (see Fig. 1i). Below we show that the transmission around sharp corners remains robust nevertheless, with this imperfection only reducing the relevant bandwidth.

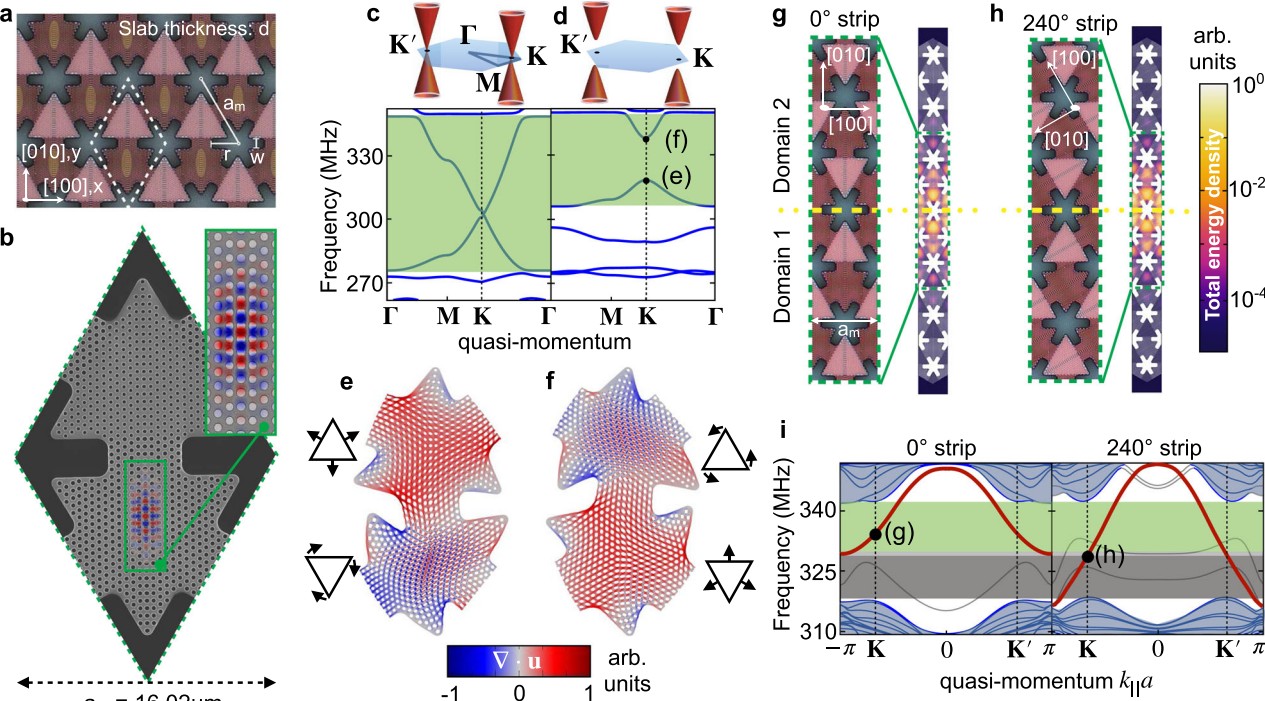

**Fig. 1 Design of the multiscale optomechanical crystal for topological phononics. a** Optical microscope image showing the snowflake triangular lattice (unit-cell dashed) with parameters $(d, r, w, a_m) = (0.22, 5.77, 2.34, 16.02)\mu m$. The axes are aligned with the silicon crystal. **b** Focused Ion Beam (FIB) image of unit-cell geometry with the simulated photonic-crystal cavity mode profile ($E_{[100]}$ component of the electric field; red/blue indicates sign). **c, d** Simulated phononic band structures with $M_y$ mirror-symmetry intact and broken (design in **b**), respectively. Inset: Sketches of the Dirac cones. **e, f** Snapshots of the mechanical mode deformation (colours indicate the local volume change, $\nabla \cdot \mathbf{u}$; red corresponding to expansion and blue compression). The arrows in the associated pictograms indicate the dynamics of the motion. **g, h** Optical microscope images and simulated mechanical mode profiles for two strip configurations, each comprising two topologically distinct domains (domain 1 as in **a**, **b**). The domain wall (dashed) has slope 0° (horizontal, **g**) or 240° (slanted, **h**) relative to the [100] axis. **i** 1-D band structures calculated for the horizontal and slanted configurations. The red lines indicate the topological edge state dispersion, the gray lines are the additional edge state modes localized at the top and bottom boundaries of the geometry (away from the domain wall), the blue parts are bulk modes. The color shading inside the bulk bandgap identifies different transport regimes in systems where the two types of domain walls are connected.

**Site-resolved optomechanical read-out of topological vibrations.** We have fabricated several devices where an internal domain of type 2 is surrounded by an external domain of type 1. The ensuing closed domain wall produces a topological mechanical cavity. In a topological cavity, counter-propagating running waves remain decoupled in spite of sharp turns and/or disorder. This gives rise to a characteristic spectrum formed by a series of doublets. These doublets are degenerate, with any slight lifting of the degeneracy due to residual inter-valley scattering.

The first topological cavity structure that we study is shown in Fig. 2a, b, consisting of an equilateral triangle of 28 snowflake unit cells along each side. A schematic of our optical setup used to measure the phononic properties of the topological cavity structure is shown in Fig. 2c. A tunable external cavity diode laser coupled to an optical fiber taper is used to optically excite individual optical nanocavities within the multiscale OMC array. The out-coupled laser light, which contains the local mechanical motion of the structure imprinted as intensity modulations, is detected on a photodiode and analyzed on an electronic spectrum analyzer. Owing to the thermal nature of the measured mechanical motion in this work, the measured electronic spectrum analyzer signal represents a local mechanical noise power spectral density (NPSD). By moving the taper position we are able to address any unit cell of the larger-scale phononic lattice, obtaining a site-resolved spectrum of the thermally populated phonon modes (see Methods for further details). As an example, we show in the top plot of Fig. 2d the resulting optically-transduced local mechanical spectrum for an optical fiber taper position at site (d) in Fig. 2a, which is in the bulk region of domain 1. The measured spectrum is seen to be in close agreement with our theoretical predictions based on FEM simulations (bottom plot of Fig. 2d), both of which show a bulk bandgap that covers an interval from 316 to 338 MHz. We note that compared to recent nanomechanical implementations[25,26], combining piezoelectric actuation and optical interferometric read-out, the displacement sensitivity in our cavity-based measurements is boosted by the cavity finesse. This has allowed us to detect tiny thermal vibrations with amplitudes on the order of 10 fm (Supplementary Note 7 and Supplementary Fig. 4).

We now focus on the domain wall region. Exploiting our single-site resolution capability, we have measured the mechanical NPSD as a function of read-out position, as shown in Fig. 2e. This reveals two dramatically different transport regimes. For the mechanical cavity modes at lower frequencies (321–327 MHz), we observe a strong modulation versus site position in each of the mechanical mode peaks. These fringe-like features indicate that thermal phonon excitations are reflected and form standing waves. This is due to the absence of topological edge modes inside the horizontal domain wall at these frequencies, resulting in standing waves inside the slanted domain wall portions of the mechanical cavity path. By contrast, we observe no such fringes in the higher-frequency regime (327–337 MHz). This indicates backscattering-immune running waves, providing a direct visual signature of the formation of a topological mechanical cavity.

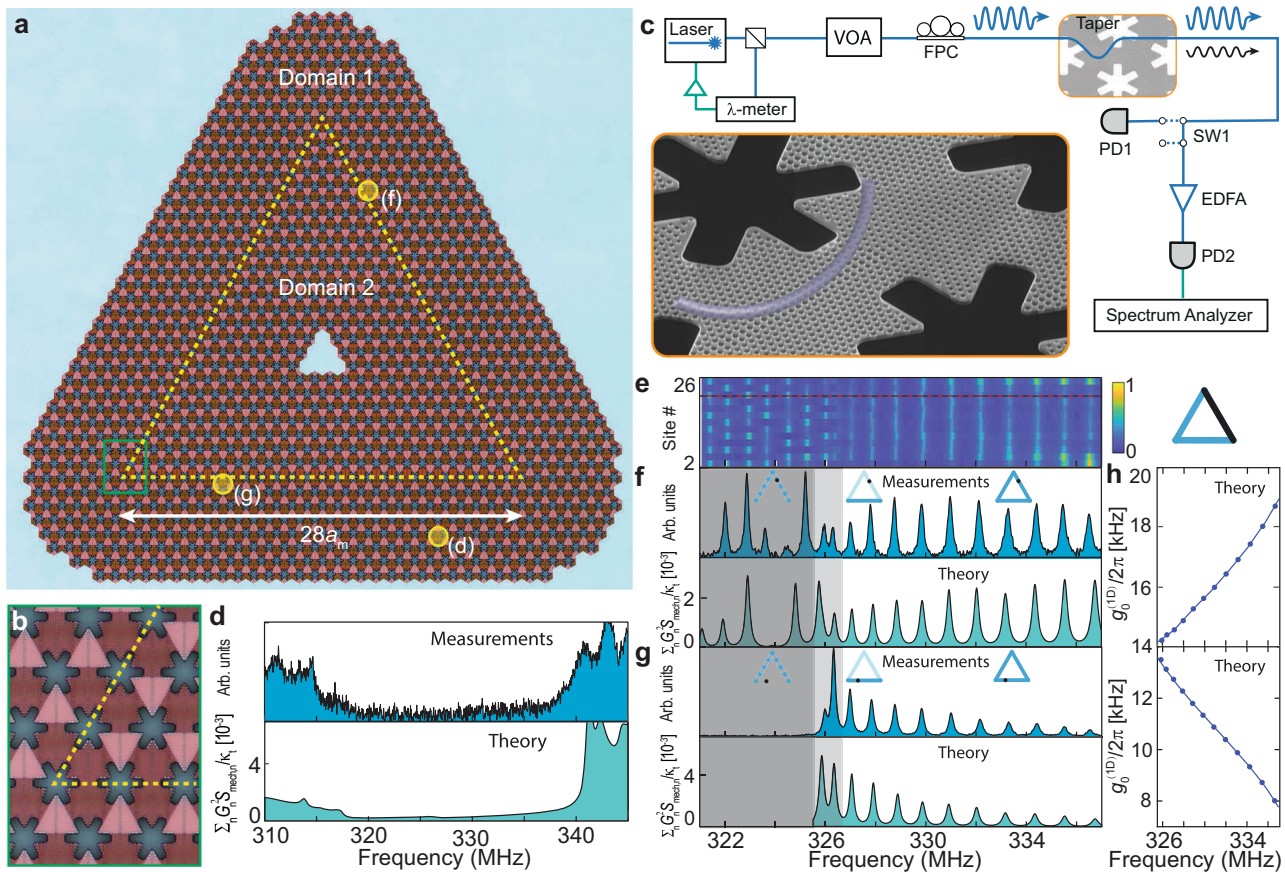

**Fig. 2 Characterization of topological edge states using optomechanical read-out. a** Optical microscope image of triangular topological mechanical cavity (Domain wall: dashed line. Read-out cavities for the measurements in **d**, **f**, and **g**: yellow dots). **b**, Zoom-in of the topological cavity corner (green box in **a**). **c** Experimental setup. Mechanical side-bands are imprinted on a laser beam transmitted through an optical cavity, detecting the NPSD of the mechanical waves. Acronyms: optical wave meter ($\lambda$ meter), variable optical attenuator (VOA), fiber polarization controller (FPC), optical switch (SW), erbium-doped fiber amplifier (EDFA), photodetector (PD). **d**, **f**, **g** Measured (top) and numerically estimated (bottom) NPSD, respectively, in the bulk of domain 1, on a slanted edge, and on a horizontal edge. Insets in **f**, and **g**: Sketches showing read-out positions and the expected local density of states. **e** Measured NPSD as a function of frequency and read-out position on a slanted edge (highlighted in black in the sketch). Red dashed line corresponds to the spectrum in **f**. The low-frequency region (dark gray in **f** and **g**) harbors modes only inside the slanted edges (cf Fig. 1i). Data calibration is required to compare measurements from different read-out cavities (see Supplementary Note 8 and Supplementary Fig. 6). **h** Optomechanical coupling for edge states in slanted (top) and horizontal (bottom) domain walls (see Supplementary Note 7).

Below, we refer to this frequency range as the topological bandwidth. In between these regimes, there is a crossover region (light gray in Fig. 2f–g), where the horizontal edge already supports edge states but backscattering is still possible because small quasi-momentum transfers are sufficient to flip right-moving into left-moving horizontal edge states due to their proximity to the Brillouin zone boundary of the horizontal edge structure (see bandstructure plot in Fig. 1i). We note that significant backscattering for edge states based on the Valley Hall effect should be expected whenever the wavefunction is not well localized within one valley. For our experiment and other setups featuring sharp domain walls, this sets a limit on the achievable topological bandwidth which could be bypassed by introducing smooth domain walls[41].

We further substantiate the absence of backscattering in the topological bandwidth by comparing the frequency dependence of the measured NPSD with theory predictions that assume perfect transmission at the corners. They are based on scattering matrix calculations that take FEM simulations as input (see Supplementary Note 7 and Supplementary Figs. 3, 5). The theoretical spectra are in good agreement with measurement results both on the slanted and the horizontal edges, as shown in Fig. 2f, g, respectively. Even the behavior of the peak heights,

distinctly different for both types of edges, is captured very well by including both the group velocity dispersion and the frequency-dependent vacuum optomechanical coupling $g_0^{(1D)}$ (see Fig. 2h) in our analysis.

**Robustness against backscattering.** While the triangle geometry is the simplest closed-loop geometry, already producing a topological mechanical cavity, we also sought to test the robustness and immunity to waveguide imperfections in more complex cavity structures where we could independently vary the length of waveguide segments between sharp corners. The effects of such variations should be most pronounced in geometry with appreciable backscattering at the corners, eventually producing separate standing wave patterns in the segments whose free spectral range would depend on the segment length. By contrast, the ideal case of robust topological transport should only be sensitive to the overall length of the domain wall circumference. Producing samples with different local geometrical details, but the same circumference, allows us to test these ideas by comparing their spectra.

To this end, we designed and fabricated two tree-shaped cavity structures. Each of these has a total domain wall circumference of

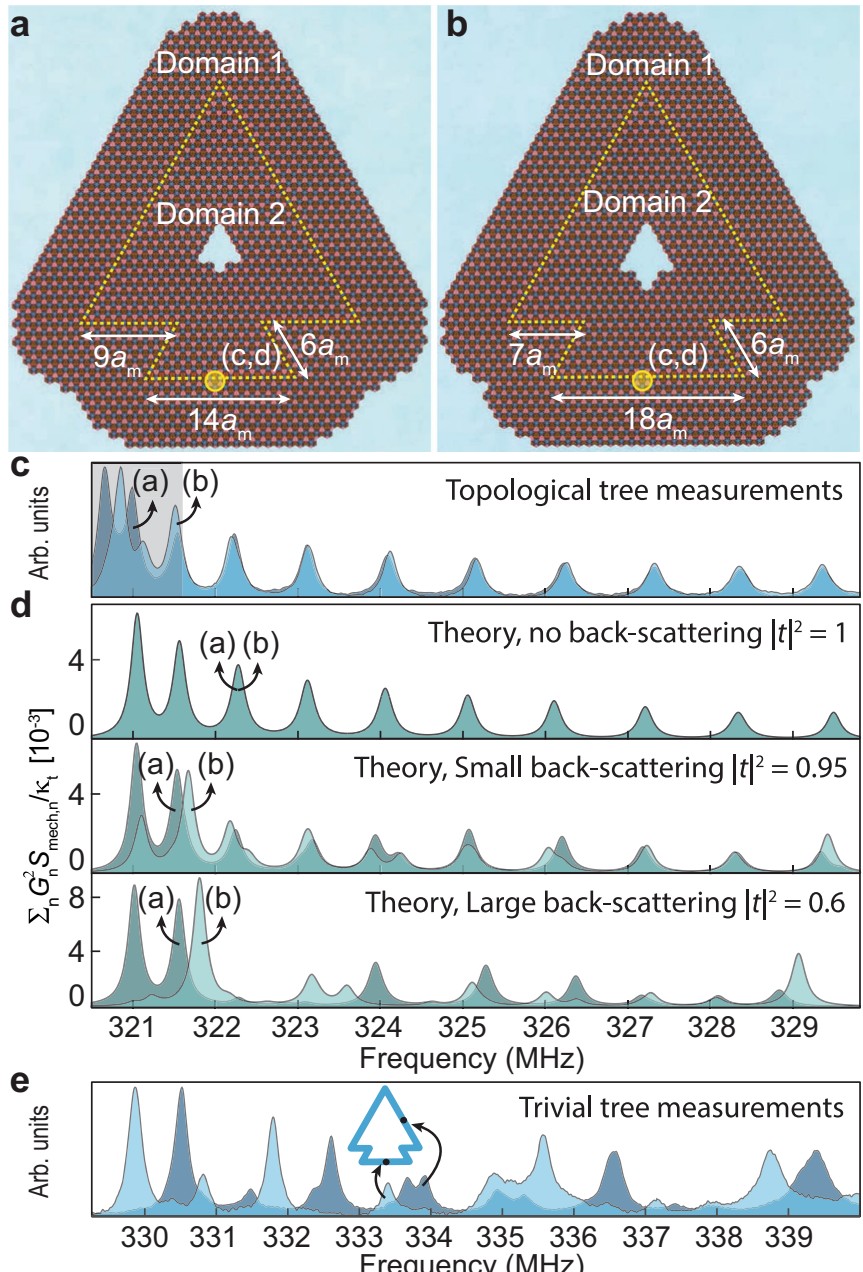

**Fig. 3 Robustness against backscattering. a, b** Optical microscope image of two different tree-shaped topological mechanical cavities. **c** Comparison of measurement results for the two trees. **d** Theoretical prediction for three backscattering strengths. Darker/lighter spectra correspond to the tree in **a/b**. In **c**, the crossover region where backscattering can occur without requiring large quasi-momentum transfer is highlighted in light gray. **e** Measurements for a trivial waveguide mechanical cavity (see Supplementary Note 10). Transduced NPSD measured at a slanted (dark blue) and a horizontal (light blue) edge, revealing strong backscattering.

96 unit cells and includes seven 60° corners, but individual segment lengths differ. Figure 3c shows the mechanical spectra measured near the horizontal edge of both tree geometries, superimposed onto each other. The most important observation is that, outside of the gray region, the two spectra agree almost perfectly, despite the different geometries. This is a clear and direct experimental signature of the near-perfect absence of backscattering, as predicted for the topological edge states. The gray region is close to the bandgap for the horizontal edge, where no suppression of backscattering is expected (see above).

In order to estimate the sensitivity of the spectra to backscattering, we performed calculations assuming varying levels of backscattering for both tree geometries (Fig. 3d), where

$|t|^2$ ($1 - |t|^2$) is the transmission (reflection) probability at each corner. These results show that even a small reflection probability of the order of 5% is enough to produce clearly visible differences between the spectra, including the splitting of the peaks. Since no such deviations are visible in the direct measurements, we conclude that the transmission surpasses 95% in our experiment, which can be seen as a figure-of-merit. This confirms that the phononic topological edge states robustly transmit through sharp corners. In addition, we have theoretically investigated the backscattering in the presence of fabrication disorder, see Supplementary Note 11 and Supplementary Fig. 8. Introducing a standard deviation of 10 nm in selected geometrical parameters of both the snowflake and the cylindrical holes, we have estimated

a round trip reflection probability $|r_{rt}|^2$ for the triangular mechanical cavity (Fig. 2) of around 1%. We expect the actual fabrication disorder using our electron-beam lithography to be in the range of 2–4 nm[42].

For further comparison, we also designed and fabricated a trivial cavity. It is created by pulling a band into the Dirac cone gap of the surrounding bulk along a line defect embedded into an otherwise uniform domain 1. In this case, the mechanical spectra measured at two different locations (on a slanted and a horizontal edge) show signatures of backscattering from the sharp corners (see Fig. 3e), with irregular peak spacing and different peak locations for the two spectra. As described in more detail in Supplementary Note 10 and Supplementary Fig. 7, the fundamental reason for larger reflections occurring in the trivial cavity is that the trivial waveguide supports both forward and backward moving modes within the same valley, greatly reducing the required quasi-momentum transfer for backscattering.

## Discussion

In conclusion, we have demonstrated a multiscale optomechanical crystal and observed topological transport of thermal phonons in the 0.3 GHz band over a bandwidth of 15 MHz. This design opens the door to implementing on-chip phononic circuits[13,15,16,43] with robust topological waveguides that have access to the full toolbox of optomechanics. Beyond cooling, mechanical lasing, sensitive read-out, and optical generation of nonclassical quantum states, this would also include the active optical control of topological circuits via local manipulation of mechanical modes (e.g., switching links between edge states). Another very promising avenue for applications consists in pushing towards even higher frequencies in the hypersonic regime — up to 100 GHz should be possible with advanced lithographic methods — inverting the scale hierarchy between photonics and phononics. This would allow one to manipulate thermal phonons in myriad of new ways, including broadband cooling of entire microscale objects, not just individual mechanical modes. Unidirectional edge channels like those found in a Chern insulator would allow one to implement thermal diodes, and, when supplemented by an energy pump, topologically protected phonon amplification and lasing[29–32]. An exciting long-term perspective is to use topological phononic circuits as the basis of a new platform to explore quantum acoustodynamics for quantum information processing and storage, with coupling to dopants or superconducting qubits.

## Methods

**Valley Hall effect: theoretical model with anisotropy.** In the Valley Hall effect, the relevant topological invariant is the so-called valley Chern number $C_v$[39,44]. The valley Chern number is defined within one valley in the framework of an effective two-band description and assumes two possible half-integer values, $C_v = \pm 1/2$. Interfaces between regions with opposite valley Chern numbers support in-gap valley-polarized edge states. Since the two valleys are mapped into each other by time reversal, their edge states are counter-propagating.

We note that due to both our elongated cavity design and the anisotropic silicon crystal (see Supplementary Note 5 and Supplementary Fig. 2), our system is not invariant under $\mathcal{C}_3$-rotations. This is a notable difference compared to previous larger-scale implementations of the Valley Hall effect[20,23,30,40,45,46]. Taking into account the residual bulk symmetry $\mathcal{T}M_x$, we find that our system is approximated by the effective two-band Dirac Hamiltonian (see Supplementary Note 4)

$$\hat{H}_D = \bar{\Omega} + (v_0 + v_x\hat{\sigma}_x)\hat{p}_x + v_y\hat{\sigma}_y\hat{p}_y + \left\{\Theta(\hat{\mathbf{r}}), \left(m + m'\hat{p}_x\right)\right\}\hat{\sigma}_z. \quad (1)$$

Here, we set $\hbar = 1$, $\hat{\sigma}_{x,y,z}$ are the Pauli matrices, {,} denotes the anti-commutator, and $\Theta(\mathbf{r}) = 1/2$ ($\Theta(\mathbf{r}) = -1/2$) inside domain 1 (domain 2). Moreover, $\mathbf{p} = (p_x, p_y)$ is the quasi-momentum counted from a point on the $k_x$-axis where the Bloch waves are mapped into each other via $M_y$, see Fig. 1e, f. The most obvious difference between $\mathcal{C}_3$-symmetric systems is that the speed of the edge state now depends on the domain wall orientation. The solutions for slanted and horizontal domain walls and other surprising features are discussed in Supplementary Note 4.

We now focus on the valley close to the **K** point. Fixing the gauge by choosing $\sigma_z = 1$ for the Bloch wave shown in Fig. 1f, a fit yields $m = 2\pi \times 10.8$ MHz, $m'/a_m = -2\pi \times 5.4$ MHz, $v_x/a_m = 2\pi \times 12.5$ MHz, and $v_y/a_m = 2\pi \times 14.9$ MHz. The valley Chern number for the lowest band is $C_v = -\text{sign}(\Theta(\mathbf{r})mv_xv_y)/2$, see Supplementary Note 4. Thus, we find $C_v = -1/2$ ($C_v = 1/2$) for domain 1 (domain 2). According to the bulk-boundary correspondence, the edge state will be a right-mover if one crosses the domain wall from domain 1 to domain 2[47]. This is consistent with our strip FEM simulations, see Fig. 1i. The expansion leading to Eq. (1) is valid if $m/a_m \ll v_y, (v_x^2 + m'^2)^{1/2}$ (see Supplementary Note 4). This condition is not strictly fulfilled in our experiment, which leads to the deviations from the ideal case remarked upon in the main text.

**Measuring the mechanical thermal fluctuations.** The thermal-mechanical motion of phonons within the multiscale OMCs of this work is measured by driving the system with the laser locked to a blue detuning of 340 MHz from the optical nanocavity resonance. This frequency offset is chosen to align with the center frequency of the mechanical Dirac cones, increasing the sensitivity of the optical read-out for phonons propagating in the topological edge states. An optical fiber taper with a localized dimple region couples light evanescently into and out of an individual optical cavity with high efficiency. By moving the taper, we can address any unit cell of the larger-scale phononic lattice. Mechanical motion is imprinted on the phase of the laser light inside the optical nanocavity, which when extracted via the optical fiber taper maps the mechanical motion into intensity modulations in the transmitted laser light. The transmitted laser signal in the optical fiber is sent through an erbium-doped fiber amplifier (EDFA) to amplify the optical intensity modulations, and then onto a high-speed photoreceiver. The RF voltage from the photoreceiver is sent into a spectrum analyzer to determine the noise power spectral density (NPSD). The NPSD of the photocurrent contains a component proportional to the sum of the mechanical NPSD $S_{\text{mech},n}$ of the mechanical normal modes of the structure, weighted by the square of the local optomechanical coupling $G_n(\mathbf{j})$, where $n$ labels the mechanical mode and $\mathbf{j}$ labels the (unit cell of the) read-out cavity (see Supplementary Note 6). Since only the vibrations within a single unit cell contribute to the optomechanical coupling $G_n(\mathbf{j})$, the transduced mechanical NPSD can be viewed as a (coarse-grained) mechanical local density of states.

## Data availability

The data supporting the results presented in this article are available at Zenodo open-access repository under [https://doi.org/10.5281/zenodo.6414313][48]. Additional data that support the findings of this study are available from the corresponding author (O.P.) upon reasonable request.

## Code availability

The code supporting the results presented in this article are available at Zenodo open-access repository under [https://doi.org/10.5281/zenodo.6414313][48].

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

## Acknowledgements
We would like to thank Sameer Sonar and Utku Hatipoglu for the help with nanofabrication and measurement. This work was supported by the Gordon and Betty Moore Foundation (award #7435) and the Kavli Nanoscience Institute at Caltech. H.R was supported by the National Science Scholarship from A*STAR, Singapore. T.S. and F.M. acknowledge support from the European Union?s Horizon 2020 research and innovation programme under the Marie Sklodowska-Curie grant agreement No. 722923 (OMT). V.P. acknowledges support by the Julian Schwinger Foundation (Grant No. JSF-16-03-0000). F.M. acknowledges support from the European Union?s Horizon 2020 Research and Innovation program under Grant No. 732894, Future and Emerging Technologies (FET)-Proactive Hybrid Optomechanical Technologies (HOT).

## Author contributions
H.R., T.S., C.B., F.M., V.P., and O.P. came up with the concept and planned the experiment. H.R., T.S., H.P., and C.B. performed the device design and fabrication. H.R. performed the measurements. H.R., T.S., F.M., V.P., and O.P. analyzed the data. All authors contributed to the writing of the manuscript.

## Competing interests
The authors declare no competing interests.
