## [Peer review file · Nature Communications]

REVIEWER COMMENTS

Reviewer #1 (Remarks to the Author):

I have read the authors' response to my earlier comments, as well as the revised manuscript. The authors have responded to all comments in a satisfactory way. I believe that the manuscript has significantly improved, with the work now put in proper context and the main points of novelty made more specific and clear. I believe the manuscript presents great work and is definitely suitable for publication in Nature Communications.

Reviewer #2 (Remarks to the Author):

In the manuscript "Topological phonon transport in an optomechanical system", Ren, Shah and co-authors report on the experimental observation of topologically protected ~ 0.3 GHz phononic modes travelling along a phononic crystal interface. Each unit cell embeds a lower-scale photonic crystal cavity enabling optomechanical sensitive spatial readout of the thermally excited phonon states all along the interface. The demonstration of topological protection is convincing, based on theoretical calculations and Finite-Element Simulations. The measurements are also compared between different designs, topological and trivial.

These results are very interesting and there is no doubt that it constitutes a significant advance at the crossroads of the fields of topological matter, phononics and optomechanics. I would therefore recommend publication in Nature Communications after the following minor points are addressed:

- While the topological protection should be robust to some level of disorder, higher disorder will unavoidably modify the band structure up to a point where the topological protection is no more present. Can the authors provide information on the tolerance of their design to disorder, regarding the topological protection?
- The experiment is realized at room temperature, but this information does not appear in the main text. I strongly suggest emphasizing this point early in the manuscript.
- In the SI, (SI-78), the authors provide information on the estimated displacement amplitude (≈ 10 fm) but what is the displacement sensitivity of the measurement setup?
- line 132-135, the authors write "we focus here on the vibrational modes that couple to light, the in-plane modes which are even under the mirror operator M_z ($z \rightarrow -z$)." Does this imply that the Comsol simulations use a symmetric condition or were the odd modes also simulated? For example, in Fig.S-2, are all the modes represented?
- Please precise the optical power launched to the cavity, in the experiments, i.e. after the polarization controller.
- It seems that the data shown in the schematic of the spectrum analyzer, Fig2.c, are not used in the discussion. Please remove if so.

Authors' Response NCOMMS-21-23583-T

January 12, 2022

Authors' Response:

We thank all the reviewers for their careful and detailed review of our manuscript. We are happy that all of them acknowledge the high quality of our work and praise it as impressive. The reviewers' points are marked in **black** while our responses are marked in **blue**.

A list of changes is given below. We hope that with these replies and the changes and additions implemented here, our manuscript is ready to be published in Nature Communications.

With best regards,
Hengjiang Ren, Tirth Shah, Hannes Pfeifer, Christian Brendel, Vittorio Peano, Florian Marquardt, Oskar Painter

Brief summary of changes

Based on the reviewers' feedback we have made the following changes in the revised version of our manuscript and the Supplemental Information:

- We have modified section 7 of the Supplementary Information to include the details about the sensitivity of the measurement setup (see reply to Referee 2).
- We have modified a paragraph in the main text and the section 11 of the Supplementary Information to highlight the robustness of the topologically protected edge state with respect to the fabrication disorder, based on new simulations (see reply to Referee 2).

Reviewer #1:

I have read the authors' response to my earlier comments, as well as the revised manuscript. The authors have responded to all comments in a satisfactory way. I believe that the manuscript has significantly improved, with the work now put in proper context and the main points of novelty made more specific and clear. I believe the manuscript presents great work and is definitely suitable for publication in Nature Communications.

[Authors' Response]: We are happy that the referee is satisfied with our responses and the improvements in the manuscript.

Reviewer #2:

In the manuscript Topological phonon transport in an optomechanical system, Ren, Shah and co-authors report on the experimental observation of topologically protected 0.3 GHz phononic modes travelling along a phononic crystal interface. Each unit cell embeds a lower-scale photonic crystal cavity enabling optomechanical sensitive spatial readout of the thermally excited phonon states all along the interface. The demonstration of topological protection is convincing, based on theoretical calculations and Finite-Element Simulations. The measurements are also compared between different designs, topological and trivial.

These results are very interesting and there is no doubt that it constitutes a significant advance at the crossroads of the fields of topological matter, phononics and optomechanics. I would therefore recommend publication In Nature Communications after the following minor points are addressed:

[Authors' Response]: We are happy that the referee thinks that our results are significant for the mentioned scientific communities. Indeed, we believe that to establish a firm connection between them is one of the main achievements of our work.

While the topological protection should be robust to some level of disorder, higher disorder will unavoidably modify the band structure up to a point where the topological protection is no more present. Can the authors provide information on the tolerance of their design to disorder, regarding the topological protection?

[Authors' Response]: The referee raises an important point about the effect of the fabrication-induced disorder on the topological protection of the edge state. Indeed, more detailed information about the disorder would be important to assess the design concept of multiscale optomechanical crystals.

Inspired by the referee's question, we have now carried out new simulations and discussed their results in a new section 11 in the revised Supplementary Information to answer this particular question by the referee, setting up a new full tight-binding simulation of the entire setup and connecting its disorder level to the underlying microscopic disorder level.

In summary, our estimates show that introducing a standard deviation of 10 nm, for geometrical parameters of both the snowflake and cylindrical holes, the round trip reflection probability $|r_{\text{rt}}|^2$ in the triangular mechanical cavity (Fig. 2 of the main-text) is around 1% - thus still quite small. We now mention this new result also in the main body of the paper.

The experiment is realized at room temperature, but this information does not appear in the main text. I strongly suggest emphasizing this point early in the manuscript.

[Authors' Response]: Thank you for pointing this out. We have modified the main text and emphasized that the experiment is realised at room temperature.

In the SI, (SI-78), the authors provide information on the estimated displacement amplitude (10 fm) but what is the displacement sensitivity of the measurement setup?

[Authors' Response]: The displacement sensitivity of the measurement setup turns out to be $S_{\text{xx}} \approx 8 \times 10^{-17} \text{ m}/\sqrt{\text{Hz}}$ ($\sqrt{\Gamma S_{\text{xx}}} \approx 90 \text{ fm}$). Thus, in a single-shot measurement (not carried out

in the present work), we would be able to determine the position (quadrature) of the mechanical oscillator up to a precision of 90 fm in a measurement time Γ^{-1} (corresponding to the decay time of the mechanical vibrations). The procedure to estimate the sensitivity is now described in the revised Supplementary Information section 7.

line 132-135, the authors write 'we focus here on the vibrational modes that couple to light, the in-plane modes which are even under the mirror operator M_z ($z \mapsto -z$).'
Does this imply that the Comsol simulations use a symmetric condition or were the odd modes also simulated? For example, in Fig.S-2, are all the modes represented?

[Authors' Response]: Yes, the Comsol simulations use an even mirror-symmetric condition in Fig.S-2. We thank the referee for pointing this out. We have now updated the caption of Fig.S-2 to describe this important point. Given the good match between theory and experiment, we conclude that any potential disorder-induced coupling between the symmetric mechanical modes of interest and antisymmetric modes turns out to be negligible (also note that the antisymmetric modes do not couple to the light).

Please precise the optical power launched to the cavity, in the experiments, i.e. after the polarization controller.

[Authors' Response]: The optical power after the polarization controller, before the dimpled fiber taper, is $240\mu\text{W}$. The optical power after the dimpled fiber taper is $60.3\mu\text{W}$. We assume the efficiencies for both sides of the dimpled fiber taper are the same, and the optical power launched to the cavity is $120.6\mu\text{W}$. We have now added this information to the Supplementary Information, section 2.

It seems that the data shown in the schematic of the spectrum analyzer, Fig2.c, are not used in the discussion. Please remove if so.

[Authors' Response]: Thank you for your suggestion. The data shown in the schematic of the spectrum analyzer are not discussed in the main text, which represents the uncalibrated raw data for Fig2.f. We have now updated Fig2.c and removed the data in the schematic of spectrum analyzer.

REVIEWERS' COMMENTS

Reviewer #2 (Remarks to the Author):

The authors have convincingly addressed all my comments and questions, and modified their manuscript accordingly. I believe this work is suitable for publication in Nature Communication.